# Inflammatory Markers Predict Blood Neurofilament Light Chain Levels in Acute COVID-19 Patients

**DOI:** 10.3390/ijms25158259

**Published:** 2024-07-29

**Authors:** Rebecca De Lorenzo, Nicola I. Loré, Annamaria Finardi, Alessandra Mandelli, Federico Calesella, Mariagrazia Palladini, Daniela M. Cirillo, Cristina Tresoldi, Fabio Ciceri, Patrizia Rovere-Querini, Angelo A. Manfredi, Mario G. Mazza, Francesco Benedetti, Roberto Furlan

**Affiliations:** 1Division of Immunology, Transplantation and Infectious Diseases, IRCCS Ospedale San Raffaele, 20132 Milan, Italy; delorenzo.rebecca@hsr.it (R.D.L.); lore.nicolaivan@hsr.it (N.I.L.); rovere.patrizia@hsr.it (P.R.-Q.); manfredi.angelo@hsr.it (A.A.M.); 2Faculty of Medicine, Università Vita-Salute San Raffaele, 20132 Milan, Italy; ciceri.fabio@hsr.it; 3Institute of Experimental Neurology, Division of Neuroscience, IRCCS Ospedale San Raffaele, 20132 Milan, Italy; finardi.annamaria@hsr.it (A.F.); mandelli.alessandra@hsr.it (A.M.); furlan.roberto@hsr.it (R.F.); 4Faculty of Psychology, Università Vita-Salute San Raffaele, 20132 Milan, Italy; calesella.federico@hsr.it (F.C.); palladini.mariagrazia@hsr.it (M.P.); 5Psychiatry and Clinical Psychobiology, Division of Neuroscience, IRCCS Ospedale San Raffaele, 20132 Milan, Italy; mazza.mariogennaro@hsr.it; 6Emerging Bacterial Pathogens Unit, IRCCS Ospedale San Raffaele, 20132 Milan, Italy; cirillo.daniela@hsr.it; 7Hematology and Bone Marrow Transplant, IRCCS Ospedale San Raffaele, 20132 Milan, Italy; tresoldi.cristina@hsr.it

**Keywords:** COVID-19, neurofilament light chain, inflammatory markers, neuroCOVID

## Abstract

Acute coronavirus disease 2019 (COVID-19) is paralleled by a rise in the peripheral levels of neurofilament light chain (NfL), suggesting early nervous system damage. In a cohort of 103 COVID-19 patients, we studied the relationship between the NfL and peripheral inflammatory markers. We found that the NfL levels are significantly predicted by a panel of circulating cytokines/chemokines, including CRP, IL-4, IL-8, IL-9, Eotaxin, and MIP-1ß, which are highly up-regulated during COVID-19 and are associated with clinical outcomes. Our findings show that peripheral cytokines influence the plasma levels of the NfL, suggesting a potential role of the NfL as a marker of neuronal damage associated with COVID-19 inflammation.

## 1. Introduction

The severe acute respiratory syndrome coronavirus-2 (SARS-CoV-2) has a broad spectrum of clinical manifestations ranging from asymptomatic infection to life-threatening multi-organ disease [1]. After the acute phase of coronavirus disease 2019 (COVID-19), physical sequelae have been observed [2]. Neurological and neuropsychiatric sequelae substantially contribute to the post-acute burden of diseases associated with COVID-19 [2]. In this context, in the first months after recovery from the acute phase, psychopathological symptomatology, cognitive impairment, chronic fatigue, and neurological symptoms were found to be strictly associated with systemic inflammation during acute disease and its pattern of change over time [3,4,5,6,7].

MRI studies confirmed that the brain suffers during acute COVID. Beyond abnormalities in the olfactory system, systematic reviews showed diffuse cerebral white matter (WM) hypodensities/hyperintensities and the involvement of the prefrontal, anterior cingulate, and insular cortex [8,9]. Peripheral markers of systemic inflammation still predict a WM microstructure several months after recovery, with more severe inflammation during the acute phase (higher serum C-reactive protein (CRP) levels and leucocyte counts) associating with reduced axial diffusivity at diffusion tensor imaging, suggesting subtle changes in the microtubular structure of axons, which might underpin the neuropsychiatric consequences of COVID-19 [10]. The search for the biomarkers of neuronal damage, able at patient admission to direct clinical choices, would be beneficial to set priorities and optimize interventions to prevent disease sequelae.

The neurofilament light chain (NfL) is a cytoskeletal intermediate filament protein of central and peripheral neurons [11], whose detection in blood has been validated as a nervous system damage biomarker in a variety of neurological diseases [12]. We showed that, during acute COVID-19, plasma NfL levels were (i) significantly increased; (ii) correlated with markers of COVID-19 severity, such as CRP, degree of respiratory insufficiency, lymphocyte and neutrophil absolute counts, and neutrophil to lymphocyte ratio; and (iii) predicted clinical outcomes [13].

Considering that systemic inflammation during acute COVID-19 predicts the neural consequences of COVID-19 and involves a robust increase in circulating cytokines, some of which are associated with neuronal insult, the aim of the present study is to define the relationship between circulating NfL levels and a panel of peripheral inflammatory biomarkers.

## 2. Results

Clinical and demographic characteristics of the cohort, divided according to sex, are resumed in Table 1. Males were marginally older than females and had significantly higher levels of CRP and marginally lower levels of IL-1ß. Males needed hospitalization and the intensive care unit (ICU) more often than females, suggesting a more severe COVID-19 acute presentation. ijms-25-08259-t001_Table 1Table 1Clinical and demographic characteristics of the patients, divided according to sex, and levels of significance of the observed differences. *p*-value < 0.05 are shown in bold.
Males (n = 63)Females (n = 40)t or χ2*p*Age (mean ± SD)59.82 ± 11.2655.16 ± 13.811.8710.064Ethnicity



African (Yes–%)2–3.17%0–0%6.0800.108Asian (Yes–%)0–0%3–7.5%European (Yes–%)52–82.54%31–77.5%South African (Yes–%)9–14.29%6–15%%Hypertension (Yes–%)28–44.44%10–25%3.9730.056Diabetes mellitus (Yes–%)16–25.4%5–12.50%3.1100.211IRC (Yes–%)3–4.76%1–2.5%0.6860.709Hospitalization (Yes–%)52–82.54%20–50%12.312**0.001**Intensive care unit (Yes–%)28–44.44%6–15%9.592**0.002**CRP (mean ± SD)129.03 ± 87.7269.53 ± 89.583.328**0.001**IL-1ß (mean ± SD)3.27 ± 3.304.92 ± 5.431.9260.057IL-1Ra (mean ± SD)4869.62 ± 18746.301454.14 ± 2286.211.1450.255IL-4 (mean ± SD)3.85 ± 2.314.16 ± 3.830.5140.609IL-6 (mean ± SD)63.99 ± 86.2540.00 ± 78.161.4260.157IL-7 (mean ± SD)52.67 ± 35.5163.23 ± 74.090.9710.334IL-8 (mean ± SD)17.71 ± 19.1915.55 ± 13.140.6270.532IL-9 (mean ± SD)649.45 ± 431.18666.58 ± 464.330.1910.849IL-10 (mean ± SD)3.05 ± 3.433.51 ± 5.300.5280.598IL-13 (mean ± SD)1.40 ± 1.552.19 ± 3.641.5100.134IL-17 (mean ± SD)30.74 ± 19.9530.25 ± 23.080.1150.909Eotaxin (mean ± SD)58.65 ± 47.2163.04 ± 65.180.3960.693FGF basic (mean ± SD)54.87 ± 39.3554.96 ± 45.450.0100.992IFN-γ (mean ± SD)49.39 ± 151.1818.62 ± 22.401.2760.205IP-10 (mean ± SD)11,223.45 ± 21578.898090.65 ± 13,122.560.8260.411MCP-1 (mean ± SD)118.61 ± 237.9175.29 ± 101.401.0890.279PDGF-BB (mean ± SD)1986.66 ± 2683.741226.82 ± 1705.521.5960.114MIP-1ß (mean ± SD)510.63 ± 336.21514.24 ± 363.440.0510.959RANTES (mean ± SD)13,159.83 ± 13,267.8719,351.32 ± 51,598.160.9090.366TNF-α (mean ± SD)175.39 ± 80.02177.41 ± 99.370.1130.910IL-18 (mean ± SD)1061.99 ± 1034.791165.16 ± 1295.010.4470.656NfL (mean ± SD)31.84 ± 33.2424.33 ± 23.731.2410.217Correlation analyses demonstrated that NfLs were correlated with PCR and Eotaxin (Table 2).
ijms-25-08259-t002_Table 2Table 2Correlation matrix of circulating cytokines/chemokines and neurofilaments. *p*-value < 0.05 are shown in bold.
PCRIL-4IL-8IL-9EotaxinMIP-1ßNFL**PCR**
0.02370.31260.0443−0.01760.06270.2453

*p* = 0.812***p* < 0.001***p* = 0.657*p* = 0.860*p* = 0.529***p* = 0.013****IL-4**0.0237
0.44320.69490.58550.67950.0725
*p* = 0.812
***p* < 0.001*****p* < 0.001*****p* < 0.001*****p* < 0.001***p* = 0.466**IL-8**0.31260.4432
0.30250.46970.28640.0499
***p* < 0.001*****p* < 0.001**
***p* = 0.002*****p* < 0.001*****p* = 0.003***p* = 0.617**IL-9**0.04430.69490.3025
0.63470.99130.0981
*p* = 0.657***p* < 0.001*****p* = 0.002**
***p* < 0.001*****p* = 0.00***p* = 0.324**Eotaxin**−0.01760.58550.46970.6347
0.64240.2638
*p* = 0.860***p* < 0.001*****p* < 0.001*****p* < 0.001**
***p* < 0.001*****p* = 0.007****MIP-1ß**0.06270.67950.28640.99130.6424
0.1019
*p* = 0.529***p* < 0.001*****p* = 0.003*****p* < 0.001*****p* < 0.001**
*p* = 0.306**NFL**0.24530.07250.04990.09810.26380.1019

***p* = 0.013***p* = 0.466*p* = 0.617*p* = 0.324***p* = 0.007***p* = 0.306



As expected, and in accordance with our previous findings [13], the NfLs were higher in hospitalized patients, compared to patients treated at home (28.109 ± 33.468 vs. 18.374 ± 32.049; *p* < 0.001), and in patients needing ICU admission than other patients (35.289 ± 27.790 vs. 25.789 ± 30.752; *p* < 0.025). The distribution of NfL was not normal (Kolmogorov–Smirnov d = 0.218, *p* < 0.01); the best fitting was observed for gamma distribution (d = 0.128) (Figure 1A). Therefore, non-parametric statistics were used to calculate the effects of predictors on NfL levels.

MARSplines ML selected a panel of factors which affected NfL plasma levels. The model identified a main effect of sex and selected (i) age, CRP, IL-4, IL-8, IL-9, Eotaxin, and MIP-1ß, as important predictors in males (adjusted R^2^ = 0.61) and (ii) CRP in females (adjusted R^2^ = 0.25). The common factors, age and CRP, showed a positive correlation with the NfLs in both groups (Age: Spearman’s Rho = 0.540, *p* < 0.0001; CRP: R = 0.451, *p* < 0.0001) (Figure 1B,C).

A homogeneity of slopes regression confirmed a significant effect of the selected biomarkers in predicting blood NfL levels, either alone as the main effect (CRP: W^2^ = 211.69, *p* < 0.0001; IL-8: W^2^ = 253.20, *p* < 0.0001; IL-9: W^2^ = 5.46, *p* = 0.0195; Eotaxin: W^2^ = 25.94, *p* < 0.0001; MIP-1ß: W^2^ = 6.85, *p* = 0.0089), or interacting with sex (IL4: W^2^ = 320.07, *p* < 0.0001).

However, an inter-correlation of the predictors was observed (Table 2), and multicollinearity diagnosed the following: VIF was higher than 1 for all factors (CRP = 1.23, IL-4 = 2.28, IL-8 = 1.68, IL-9 = 64.16, Eotaxin = 2.20, and MIP-1ß = 64.04), yielding an average VIF substantially > 1 (VIF = 22.6), which indicates that predictors have linear relationships among themselves that might bias regression [14]. The PCA significantly identified two components that cumulatively explained 76.01% of the variance (Table 3). Only the first component, to which all predictors contributed positively, had an eigenvalue > 1 and a high Q^2^. The score of the first component was then extracted and used as an independent factor to test the combined effect of the original collinear predictors of the NfL [15].

A GLZM analysis with age and PCA scores as factors and NfL plasma levels as the dependent variable showed a significant effect of both age (older age, higher NfL; LR χ^2^ = 5.36, *p* = 0.0206) and PCA scores (higher cytokines, higher NfL; LR χ^2^ = 16.85, *p* < 0.0001), thus confirming a significant effect of peripheral inflammatory markers in predicting NfL plasma levels. A separate-slope regression showed a significant interaction of PCA scores with sex in predicting the NfL (LR χ^2^ = 6.425, *p* = 0.0403), with parameter estimates higher in males than in females (b = 3.41 vs. b = 2.38, respectively) (Figure 1D).

## 3. Discussion

Our results showed that peripheral levels of circulating cytokines/chemokines predicted plasma levels of NfL during acute COVID-19. In accordance with the available literature, this effect was found to be stronger in males, who also showed higher levels of CRP. Significant differences in immune responses during the progression of SARS-CoV-2 infection were identified between males and females, with males showing an intensified innate immune response and females demonstrating a more robust adaptive immune response. These sex-specific immune response variations suggest potential immunological mechanisms underlying the distinct disease progression pathways between the sexes, highlighting the importance of considering sex-dependent strategies in prevention and treatment [16,17].

When directly looking at the specific cytokines found to be able to predict the NfL levels (CRP, IL-4, IL-8, IL-9, Eotaxin, and MIP-1ß), the literature highlight that all the selected factors are highly up-regulated during COVID-19 and associated with clinical outcomes. In detail, systematic reviews and meta-analyses associated the excessive and uncontrolled release of IL-4, IL-8 [18], IL-9 [19], and Eotaxin [20] with more severe COVID-19. MIP-1β was associated with survival [21], but higher levels predicted a persistent inflammatory phenotype in patients with severe disease [22]. MIP-1β is a chemotactic factor for different cell types, but especially NK cells and monocytes [23]. IL-8 is a chemokine that mostly attracts and activates neutrophils, considered crucial in neurodegeneration [24]. IL-9 is a pleiotropic cytokine expressed by many different cell types and is a determinant of bronchial hyper-responsiveness [25]. IL-4 and eotaxin are typical Th2 markers crucial for eosinophils activation and attraction [26]. Taking together all the selected cytokines, it appears that both innate inflammatory response (CRP, IL-8, and MIP-1β) and Th2 response (IL-4, IL-9, and eotaxin) mediators were associated with the NfL. Consistent with our findings, a recent study found that in a sample of 175 patients admitted with COVID-19, the NfL and GFAP were associated with elevations of pro-inflammatory cytokines and the presence of auto-antibodies [27]. The associations between the NfL and the dysregulation of immune responses suggest that the inflammatory process promoted by SARS-CoV-2 infection could potentially translate into neuroinflammation, thus causing axonal damage. Of course, our data do not define causality between the immunological parameters and the presence of brain injury and need to be considered as preliminary. Future studies need to better explore this correlation and potentially associate the laboratory findings with brain imaging and clinical aspects.

Notably, only in males, age was selected as a potential predictor of the NfL plasma level, thus suggesting that SARS-CoV-2 infection and related inflammatory response could enhance the detrimental effect of age on axonal damage.

First, the relatively small sample size and the monocentric nature of the study suggest that our results should be considered preliminary. Larger multicentric case–control studies are necessary to validate these findings. Second, the observational design of the study limits the generalizability of our results. Lastly, limited healthcare resources and the emergency setting constrained our ability to obtain brain magnetic imaging scans for this sample, preventing us from correlating inflammation, NfL levels, brain structure, and functional integrity.

These limitations, however, do not bias the main finding. Brain injury in the context of the dysregulation of both innate and adaptive immune responses is a common consequence of COVID-19. In light of these data and of our results of increased NfL levels in severe COVID-19, it is tempting to speculate that the axonal damage measured by the NfLs could be associated, possibly caused, by the inappropriate activity of innate immune cells, such as neutrophils, eosinophils, NK cells, and monocytes. Further research is needed to verify this hypothesis.

## 4. Material and Methods

This retrospective cohort investigation included 103 patients (63 males, 40 females), aged ≥ 18 years, enrolled in the COVID-BioB study, a comprehensive observational study conducted at the San Raffaele University Hospital. The patients had been admitted at the emergency department (ED) of our institution for COVID-19 between 18 March and 5 May 2020 and received the diagnosis of COVID-19 based on a positive SARS-CoV-2 real-time reverse-transcriptase polymerase chain reaction (RT-PCR) from a nasopharyngeal swab and in the presence of clinical and/or radiologic findings of COVID-19 pneumonia. As part of the COVID-BioB protocol, blood samples from all patients were collected at ED arrival during acute disease. Median (interquartile range) time from ED admission to venipuncture was 1 (0–2) days. Blood samples were stored in a dedicated biobank at our institution according to appropriate quality control strategies [28]. The plasma-EDTA was obtained by centrifugation of venous blood, immediately frozen and maintained at −80 °C until subsequent analyses. The study protocol was approved by the Hospital Ethics Committee (protocol no. 34/int/2020) and registered on ClinicalTrials.gov (NCT04318366). It was conducted in accordance with the Declaration of Helsinki, and all patients signed informed consent.

The CRP was measured in plasma as a standard of care at ED presentation. The plasma was inactivated using tri-(n-butyl) phosphate and Triton X-100 (Sigma-Aldrich, Saint Louis, MO, USA) (0.3% and 1%, respectively) for 2 h [29]. The NfLs were measured with a Simoa Human Neurology 4-Plex B assay (N4PB) on a Quanterix SIMOA HD-1 platform (Quanterix, Billerica, MA, USA) according to the manufacturer’s instructions [30]. Multiplex immunoassays based on Luminex technology (Bio-Rad Laboratories, Hercules, CA, USA) were used for the quantification of 27 biomarkers among cytokines, chemokines and growth factors, according to the manufacturer’s instructions (Bio-Plex Pro™ Human Cytokine 27-plex): Interleukine (IL-) 1ß, IL-1 receptor antagonist, IL-4, IL-6, IL-7, IL-8, IL-9, IL-10, IL-13, IL-17, Eotaxin, basic fibroblast growth factor (FGF), Interferon γ (IFN-γ), Interferon gamma-induced protein 10 (IP-10), monocyte chemoattractant protein 1 (MCP-1), platelet-derived growth factor subunit B (PDGF-BB), macrophage inflammatory protein-1 beta (MIP-1ß), regulated on activation normal T cell expressed and secreted (RANTES), tumor necrosis factor alpha (TNF-α), and IL-18. Data were measured on a Bio-Plex 200 System and calculated using Bio-Plex Manager 6.0 and 6.1 software.

All the statistical analyses were performed using StatSoft Statistica v12.0 and standard computational methods [31].

First, we studied the relationship between the peripheral NfL levels and the set of predictor variables by using a nonparametric regression model with multivariate adaptive regression splines (MARSplines). The MARSplines algorithms were presented by Friedman as a method for the flexible regression modeling of high dimensional data. It has more power and flexibility to model relationships that are nearly additive or involve interactions in, at most, a few variables. This data-driven supervised machine learning (ML) approach allows for parsimoniously selecting the best subset of predictors by evaluating sub-models and selecting the best one without assuming any particular type of relationship among variables and using a pruning technique to boost model sparsity and, thus, constrain the complexity of the model [32]. MARSplines has been proven useful for feature selection and data reduction when analyzing heterogeneous medical datasets [33,34], as related either with infectious diseases, cancer, gender medicine, public health, electrocardiographic imaging, or molecular biology [35,36,37,38,39,40]. We entered sex, age, and plasma levels of CRP, IL-1ß, IL-1Ra, IL-4, IL-6, IL-7, IL-8, IL-9, IL-10, IL-13, IL-17, Eotaxin, FGF basic, IFN-γ, IP-10, MCP-1, PDGF-BB, MIP-1ß, RANTES, TNF-α, and IL-18 as a panel of factors to predict the peripheral levels of NfL. The factors were then sequentially eliminated from further analysis if determined to be nonsignificant contributors to the model.

Second, we aimed at confirming the significant effect of the selected predictors on the NfLs by performing homogeneity of slopes or separate-slopes regressions as appropriate, in the frame of the generalized linear model (GLZM) [41]. The parameter estimates were obtained with iterative re-weighted least squares maximum likelihood procedures. The significance of the effects was calculated with the Wald W2 statistics, or with likelihood ratio (LR) statistic, which provides the most asymptotically efficient test known, by performing sequential tests for the effects in the model of the factors on the dependent variable, at each step adding an additional effect into the model contributing to incremental χ2 statistic, thus providing a test of the increment in the log-likelihood attributable to each current estimated effect [42,43].

Multicollinearity was tested by calculating the variance inflation factor (VIF). In order to address multicollinearity, a principal component analysis (PCA) was run with data whitening and feature extraction purposes [44,45]. The PCA was performed to identify the orthogonal directions of maximum variance in the original data and to project the data into a lower-dimensionality space formed of a subset of the highest-variance component(s), detected according to the least square criterion.

## Figures and Tables

**Figure 1 ijms-25-08259-f001:**
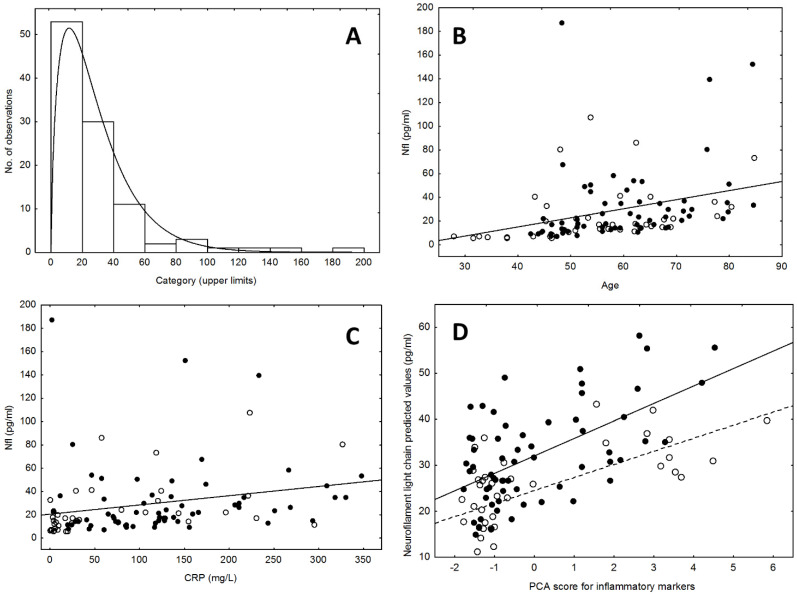
(**A**) Distribution of blood NfL levels among participants and fitting of gamma distribution. (**B**) Relationship between plasma CRP and NfL levels (black dots: males; white dots: females). (**C**) Relationship between age and NfL levels (black dots: males; white dots: females). (**D**) Relationship between peripheral inflammatory markers (PCA scores) and predicted plasma NfL levels (black dots: males; white dots: females).

**Table 3 ijms-25-08259-t003:** Principal components analysis results.

	Component 1 R^2^ = 0.562; Eigenvalue = 3.37; Q^2^ = 0.419	Component 2 R^2^ = 0.198; Eigenvalue = 0.76; Q^2^ = 0.021
CRP	0.122076	0.875180
IL-4	0.837744	−0.053015
IL-8	0.563676	0.587793
IL-9	0.918200	−0.192722
Eotaxin	0.810904	−0.056926
MIP-1ß	0.914190	−0.186649

## Data Availability

The data presented in this study are available on request from the corresponding author due to privacy reasons.

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
