# Peer review of "Inflammatory Markers Predict Blood Neurofilament Light Chain Levels in Acute COVID-19 Patients"

_ijms, 2024, doi:10.3390/ijms25158259_

Round 1

Reviewer 1 Report

Comments and Suggestions for Authors

The COVID-19 pandemic has prompted extensive research into the mechanisms of the disease and its impact on various bodily systems. Among the many biomarkers investigated, Neurofilament light chain (NfL) and peripheral cytokines have emerged as significant indicators of neurological involvement and systemic inflammation, respectively, in COVID-19 patients. Studies have reported elevated levels of NfL in the plasma of COVID-19 patients, particularly those with severe disease and neurological manifestations. Elevated NfL levels suggest that SARS-CoV-2 infection can cause neuronal damage, possibly due to hypoxia, inflammation, or direct viral effects on the nervous system. This manuscript studied the relationship between NfL and peripheral inflammatory markers with a cohort of 103 COVID-19 patients, they found the NfL levels are significantly predicted by circulating cytokines/chemokines, these data increased our knowledge about the NfL as a marker of neuronal damage associated with COVID-19 infection. However, there are several major and minor weaknesses in the rationale and research methods of this work. Below please find the review comments. 

(1) Major comments

1. The concern about the novelty of the study due to the previous reporting of NfL as a marker of neuronal damage associated with COVID-19 infection is noted. To address this, it is crucial to validate the correlation of circulating cytokines/chemokines, specifically CRP, IL-4, IL-8, IL-9, Eotaxin, and MIP-1ß, with NfL levels. This validation will enhance the study's contribution by exploring the interplay between these inflammatory markers and NfL, potentially shedding light on their combined role in neuronal damage during COVID-19 infection.

2, Examining the differences in NfL as a marker of neuronal damage associated with COVID-19 severity separately for males and females would indeed add valuable insights to the study. This approach could elucidate potential sex-specific differences in neurologic outcomes during COVID-19 infection, considering that males and females may exhibit varying immune responses and disease severities. Analyzing NfL levels separately by sex could help determine if there are differential patterns of neuronal damage and recovery between male and female COVID-19 patients, contributing to a more nuanced understanding of neurologic implications in this context (figure 1B).

3. It appears that the author did not consider the factor of aging to NfL concentration in the COVID cohort analysis. The contribution of aging to these circulating cytokines/chemokines, as well as NfL concentration levels should be mentioned, if COVID infection magnified (or covered) the aging effect? Please clarify this points when interpreting the data.

4. Adding detailed clinical and demographic characteristics to Table 1, including race, ethnicity, geographic region, medical history, current health status, medications, and family history, would indeed provide a comprehensive overview of the recruited patients and their potential influence on circulating cytokines/chemokines in the study. This additional information could enhance the understanding of how these factors contribute to the immune responses and outcomes observed in relation to NfL levels during COVID-19 infection.

(2) Minor comments

1. Abbreviations should be properly defined and interpreted when they are first introduced in the manuscript. This practice ensures that readers can understand the meaning of abbreviations without confusion. It is also recommended to include a comprehensive list of abbreviations after the conclusion section. This list will serve as a quick reference for readers, enabling them to easily access the definitions throughout the manuscript.

2. Line 129, Material and Methods. It is better to describe the details of the blood sample processing, i.e., the protocol from blood to plasma samples. Meanwhile, the catalog code of each kit is also required.

3. When talking about the “relationship between peripheral NfL levels and circulating cytokines/chemokines”, I believe the different feature of plasma from male and female patients should be discussed, and these literatures are essential to refer. doi.org/10.1186/s13293-021-00410-2, doi: 10.1101/2020.06.06.20123414.

4. The authors should make the necessary modifications to the format of all the tables to ensure they comply with the specific journal requirements. This may involve adjusting the table layout, font style and size, column headings, and other elements to align with the journal's guidelines.

6. Limitations of this study should be discussed objectively.

Comments on the Quality of English Language

N/A

Author Response

The COVID-19 pandemic has prompted extensive research into the mechanisms of the disease and its impact on various bodily systems. Among the many biomarkers investigated, Neurofilament light chain (NfL) and peripheral cytokines have emerged as significant indicators of neurological involvement and systemic inflammation, respectively, in COVID-19 patients. Studies have reported elevated levels of NfL in the plasma of COVID-19 patients, particularly those with severe disease and neurological manifestations. Elevated NfL levels suggest that SARS-CoV-2 infection can cause neuronal damage, possibly due to hypoxia, inflammation, or direct viral effects on the nervous system. This manuscript studied the relationship between NfL and peripheral inflammatory markers with a cohort of 103 COVID-19 patients, they found the NfL levels are significantly predicted by circulating cytokines/chemokines, these data increased our knowledge about the NfL as a marker of neuronal damage associated with COVID-19 infection. However, there are several major and minor weaknesses in the rationale and research methods of this work. Below please find the review comments. 

(1) Major comments

1. The concern about the novelty of the study due to the previous reporting of NfL as a marker of neuronal damage associated with COVID-19 infection is noted. To address this, it is crucial to validate the correlation of circulating cytokines/chemokines, specifically CRP, IL-4, IL-8, IL-9, Eotaxin, and MIP-1ß, with NfL levels. This validation will enhance the study's contribution by exploring the interplay between these inflammatory markers and NfL, potentially shedding light on their combined role in neuronal damage during COVID-19 infection.

ANSWER: according with the reviewer we included NfL levels in the table 2 in order to report the correlation between circulating cytokines/chemokines and NfL. NfL correlated with PCR and Eotaxin, we specified this finding the result section.

2, Examining the differences in NfL as a marker of neuronal damage associated with COVID-19 severity separately for males and females would indeed add valuable insights to the study. This approach could elucidate potential sex-specific differences in neurologic outcomes during COVID-19 infection, considering that males and females may exhibit varying immune responses and disease severities. Analyzing NfL levels separately by sex could help determine if there are differential patterns of neuronal damage and recovery between male and female COVID-19 patients, contributing to a more nuanced understanding of neurologic implications in this context (figure 1B).

ANSWER: we completely agree with the reviewer regarding the importance of stratifying for sex. In line with this all the descriptive analyses where performed also stratifying for the effect of sex (see table 1 and figure 1B, 1C, and 1D). Moreover according with the reviewer suggestion (point 2 and point 4) we included other relevant variables and specifically we included the setting of the cure (treated at home, hospitalization, need of ICU) as a proxy of COVID-19 severity. As expected from the epidemiology and also according to our previous paper, males shows more severe acute COVID causing higher rate of hospitalization and ICU in males. We modified the result section according with this point.

Moreover, in order to investigate the effect of COVID severity on NfL, we investigated the relationship between NfL and i) need of hospitalization; ii) need of ICU. According with literature and with our previous findings, we found that hospitalized patients and ICU patients showed higher NfL. We modified the results section accordingly and we stressed the importance of sex-different immune response in the discussion.

3. It appears that the author did not consider the factor of aging to NfL concentration in the COVID cohort analysis. The contribution of aging to these circulating cytokines/chemokines, as well as NfL concentration levels should be mentioned, if COVID infection magnified (or covered) the aging effect? Please clarify this points when interpreting the data.

ANSWER: we agree with the reviewer regarding the importance of aging when dealing wit NfL. We explored the effect of age on NfL and we found a positive correlation with NfL (Spearman's Rho=0.540, p<0.0001) (See results and Figure 1B). Considering this effect, the age was entered as covariate in the nonparametric regression modelling with multivariate adaptive regression splines (MARSplines). Among the selected predictors which affected NfL plasma levels, the MARSplines selected also age as important predictors in males. According to this point, we have now briefly discussed this point in the discussion section.

4. Adding detailed clinical and demographic characteristics to Table 1, including race, ethnicity, geographic region, medical history, current health status, medications, and family history, would indeed provide a comprehensive overview of the recruited patients and their potential influence on circulating cytokines/chemokines in the study. This additional information could enhance the understanding of how these factors contribute to the immune responses and outcomes observed in relation to NfL levels during COVID-19 infection.

ANSWER: according with this point we have included more clinical and demographical information in Table 1 in order to better describe the sample. We included ethnicity, current health status (hypertension, diabetes mellitus, chronic kidney disease) and setting of care for COVID as a proxy of acute COVID severity (see also point 2). Apart from setting of care, no other variables were found to differ between groups.

Minor comments

1. Abbreviations should be properly defined and interpreted when they are first introduced in the manuscript. This practice ensures that readers can understand the meaning of abbreviations without confusion. It is also recommended to include a comprehensive list of abbreviations after the conclusion section. This list will serve as a quick reference for readers, enabling them to easily access the definitions throughout the manuscript.

ANSWER: we modified according the suggestion and we included the list.

2.Line 129, Material and Methods. It is better to describe the details of the blood sample processing, i.e., the protocol from blood to plasma samples. Meanwhile, the catalog code of each kit is also required.

ANSWER: we better described the process from blood to plasma samples in Material and Methods section.

3.When talking about the “relationship between peripheral NfL levels and circulating cytokines/chemokines”, I believe the different feature of plasma from male and female patients should be discussed, and these literatures are essential to refer. doi.org/10.1186/s13293-021-00410-2, doi: 10.1101/2020.06.06.20123414.

ANSWER: we thanks the reviewer for the references. We have modified the discussion stressing this point

4. The authors should make the necessary modifications to the format of all the tables to ensure they comply with the specific journal requirements. This may involve adjusting the table layout, font style and size, column headings, and other elements to align with the journal's guidelines.

ANSWER: we modified the table according with journal’s guidelines (https://www.mdpi.com/journal/ijms/instructions). Tables are now in the main test near to the first time they are cited. All tables are consistent with the journal requirements: “All table columns should have an explanatory heading. To facilitate the copy-editing of larger tables, smaller fonts may be used, but no less than 8 pt. in size. Authors should use the Table option of Microsoft Word to create tables.”

5. Limitations of this study should be discussed objectively.

ANSWER: we have introduced a limitation section at the end of the discussion.

Reviewer 2 Report

Comments and Suggestions for Authors

The authors studied the relationship between NfL and peripheral inflammatory markers in 103 patients who visited the emergency department with COVID-19 and pneumonia. Although this study is unique, we cannot recommend its publication for the following reasons:

It is unclear why the association between NfL levels and inflammatory markers was studied only in COVID-19 patients. NfL levels are expected to be elevated in patients with systemic infections and inflammation of the nervous system, not limited to COVID-19. To suggest differences in the inflammatory cascade between SARS-CoV-2 and other pathogens, it may be useful to compare COVID-19 patient groups with patient groups with infections caused by other pathogens.

The same authors reported that NfL is elevated in COVID-19 patients, and citing other papers would be useful to confirm whether NfL is specifically associated with SARS-CoV-2 infection.

The discussion section mainly consists of a list of previous reports and does not provide a proper discussion of this study.

In this study, many variables were used in the multivariate analysis, and the number of subjects was only 103. The authors need to explain whether appropriate statistical analysis can be performed with this number of subjects.

Author Response

The authors studied the relationship between NfL and peripheral inflammatory markers in 103 patients who visited the emergency department with COVID-19 and pneumonia. Although this study is unique, we cannot recommend its publication for the following reasons:

It is unclear why the association between NfL levels and inflammatory markers was studied only in COVID-19 patients. NfL levels are expected to be elevated in patients with systemic infections and inflammation of the nervous system, not limited to COVID-19. To suggest differences in the inflammatory cascade between SARS-CoV-2 and other pathogens, it may be useful to compare COVID-19 patient groups with patient groups with infections caused by other pathogens.

ANSWER: this was not the aim of our study. We know that NfL may be elevated in several neuropsychiatric inflammatory, degenerative or infective disease as a biomarker of axonal damage. Regarding COVID, our group have already published a paper were blood neurofilament light chain and total tau levels at admission were found to predict death in COVID-19 patients. In the present paper, given this a priori, we wanted to investigate the pathophysiological relationship between NfL and systemic inflammation in COVID survivors as clearly state in the introduction “aim of the present study is to define the relationship between circulating NfL levels and a panel of peripheral inflammatory biomarkers.

The same authors reported that NfL is elevated in COVID-19 patients, and citing other papers would be useful to confirm whether NfL is specifically associated with SARS-CoV-2 infection.

ANSWER: we don’t think that this mechanism is specific for COVID and we have never stated this in the manuscript. We think that COVID related inflammation could affect axonal damage thus representing a pathophysiological mechanism able to explain the relevant neuropsychiatric consequences of SARS-CoV-2 infection. We consider the SARS-CoV-2 infection as a trigger (as well as other infection, MS relapses, brain hipoxia and so on) able to induce systemic inflammation that translate into neuroinflammation thus causing axonal damage resulting in higher NfL. We discussed our hypothesis accordingly:

“Brain injury in the context of dysregulation of both innate and adaptive immune responses is a common consequence of COVID-19. In light of these data and of our results of increased NfL levels in severe COVID-19, it is tempting to speculate that the axonal damage measured by NfL could be associated, possibly caused, by the inappropriate activity of innate immune cells, such as neutrophils, eosinophils, NK cells and monocytes”.

The discussion section mainly consists of a list of previous reports and does not provide a proper discussion of this study.

ANSWER: we submitted the present paper as a short communication considering the relative small sample size. Given the format we have summarized the discussion as much as possible in order to have enough words to completely describe the sample and the methods used.

In this study, many variables were used in the multivariate analysis, and the number of subjects was only 103. The authors need to explain whether appropriate statistical analysis can be performed with this number of subjects.

ANSWER: the statistical methodology is appropriate and clearly explained in the “Methods” section. I report the description that, in our opinion, clarify the reviewer doubts:

This data-driven supervised Machine Learning (ML) approach allows for parsimoniously selecting the best subset of predictors by evaluating submodels and select the best one without assuming any particular type of relationship among variables, and using a pruning technique to boost model sparsity and thus constrain the complexity of the model. MARSplines has been proven useful for feature selection and data reduction when analyzing heterogeneous medical datasets as related either, with infectious diseases, cancer, gender medicine, public health, electrocardiographic imaging, or molecular biology. We entered sex, age, and plasma levels of CRP, IL-1ß, IL-1Ra, IL-4, IL-6, IL-7, IL-8, IL-9, IL-10, IL-13, IL-17, Eotaxin, FGF basic, IFN-γ, IP-10, MCP-1, PDGF-BB, MIP-1ß, RANTES, TNF-α, and IL-18, as a panel of factors to predict peripheral levels of NfL. Factors were then sequentially eliminated from further analysis if determined to be nonsignificant contributors to the model.

Round 2

Reviewer 1 Report

Comments and Suggestions for Authors

The authors have revised the paper with addition of new data and have addressed all the point raised in the initial review.

Author Response

We thanks the reviewer. 

Reviewer 2 Report

Comments and Suggestions for Authors   Thank you for your reply. I understood the purpose of this study. However, I would emphasize again that the total number of subjects in this study is insufficient to perform the analysis using multivariate adaptive regression splines. Having too many variables in the analysis increases the frequency of false significance. The author's response does not solve this problem.   This study found that IL-4, 8, 9, Eotaxin, and MIP-1β affect plasma levels of elevated NfL. However, the functions of these cytokines are diverse. The question of why these cytokines affect NfL has not been fully investigated. We feel that simply creating statistical models is not sufficient for medical research.

Author Response

1.Thank you for your reply. I understood the purpose of this study. However, I would emphasize again that the total number of subjects in this study is insufficient to perform the analysis using multivariate adaptive regression splines. Having too many variables in the analysis increases the frequency of false significance. The author's response does not solve this problem.

1. ANSWER We cannot agree with the reviewer. We think that the statistical pipeline perfectly fit with the data. MARSplines is very useful to solve high-dimensional problems, such as a large number of inputs and does not require assumptions about the functional relationship between independent (input) and dependent (output) data. In this context, we underline that the multivariate adaptive regression splines (MARSplines) were presented by Friedman as a method for flexible regression modelling of high dimensional data.

Literally “A new method is presented for flexible regression modelling of high dimensional data. The model takes the form of an expansion in product spline basis functions, where the number of basis functions as well as the parameters associated with each one (product degree and knot locations) are automatically determined by the data. This procedure is motivated by the recursive partitioning approach to regression and shares its attractive properties. Unlike recursive partitioning, however, this method produces continuous models with continuous derivatives. It has more power and flexibility to model relationships that are nearly additive or involve interactions in at most a few variables. In addition, the model can be represented in a form that separately identifies the additive contributions and those associated with the different multivariable interactions.” (Friedman, J. H. (1991). Multivariate adaptive regression splines. The annals of statistics, 19(1), 1-67).

To solve any doubt, and to better clarify the issue we also modified the manuscript (Methods section) to stress this point.

Moreover, as reported in the manuscript, MARSplines procedure was used for the development of predictive models in different medical field and was chosen as a promising tool for prediction models:

  1. Koc EK, Bozdogan H. Model selection in multivariate adaptive regression splines (MARS) using information complexity as the fitness function. Machine Learning. 2015;101(1):35-58.
  2. Kumar DS, Sukanya S, BIT-Campus T. Feature selection using multivariate adaptive regression splines. International Journal of Research and Reviews in Applied Sciences And Engineering (IJRRASE). 2016;8(1):17-24.
  3. Put R, Vander Heyden Y. The evaluation of two‐step multivariate adaptive regression splines for chromatographic retention prediction of peptides. Proteomics. 2007;7(10):1664-77.
  4. Chou S-M, Lee T-S, Shao YE, Chen I-F. Mining the breast cancer pattern using artificial neural networks and multivariate adaptive regression splines. Expert systems with applications. 2004;27(1):133-42.
  5. York TP, Eaves LJ, van den Oord EJ. Multivariate adaptive regression splines: a powerful method for detecting disease–risk relationship differences among subgroups. Stat Med. 2006;25(8):1355-67.
  6. Xu Q-S, Daszykowski M, Walczak B, Daeyaert F, De Jonge M, Heeres J, et al. Multivariate adaptive regression splines—studies of HIV reverse transcriptase inhibitors. Chemom Intell Lab Syst. 2004;72(1):27-34.
  7. Onak ON, Erenler T, Serinagaoglu Y. A Novel Data-Adaptive Regression Framework Based on Multivariate Adaptive Regression Splines for Electrocardiographic Imaging. IEEE Trans Biomed Eng. 2021.
  8. Martín Cervantes PA, Rueda López N, Cruz Rambaud S. The Relative Importance of Globalization and Public Expenditure on Life Expectancy in Europe: An Approach Based on MARS Methodology. Int J Environ Res Public Health. 2020;17(22):8614.

Finally, to be as conservative as possibile, we confirmed the significant effect of the selected predictors on NfL by performing homogeneity of slopes or separate-slopes regressions as appropriate, in the frame of the generalized linear model.

For abovementioned reasons, we think that the MARSplines algorithm is a good choice to deal with our data and sample.

2.This study found that IL-4, 8, 9, Eotaxin, and MIP-1β affect plasma levels of elevated NfL. However, the functions of these cytokines are diverse. The question of why these cytokines affect NfL has not been fully investigated. We feel that simply creating statistical models is not sufficient for medical research. The discussion section mainly consists of a list of previous reports and does not provide a proper discussion of this study.

2. ANSWER: Considering the nature of the trigger (a respiratory virus), not surprisingly, the cytokines found to affect NfL are all involved in innate inflammatory response (CRP, IL-8 and MIP-1β) and Th2 response (IL-4, IL-9 and eotaxin). These cytokines were found to be all highly up-regulated during COVID-19 (from the manuscript: “literature highlight that all the selected factors are highly up-regulated during COVID-19 and associate with clinical outcomes. In detail, systematic reviews and meta-analyses as-sociated the excessive and uncontrolled release of IL-4, IL-8 [36], IL-9 [37], and Eotaxin [38] with more severe COVID-19”). Of course, our data do not define causality between the immunological parameters and the presence of brain injury and need to be considered as preliminary. Future studies need to better explore this correlation and potentially associate the laboratory findings with brain imaging and clinical aspects.

According to this point we have modified the discussion to briefly (according with the communication format) discuss this issue and to clearly state that our data do not define causality between the immunological parameters and the presence of brain injury.

Round 3

Reviewer 2 Report

Comments and Suggestions for Authors

Thank you for revising the manuscript. I have no additional comments to add.